# Investigating the Impact of Different Partial Overlap Levels on the Perception of Visual Variables for Categorical Data

**Diego Santos** *,† , **Alexandre Freitas** , **Rodrigo Lima** , **Carlos Gustavo Santos** and **Bianchi Meiguins** †

Computer Science Postgraduate Program, Federal University of Pará, Belém 66075-110, Brazil; alexandre.freytas@gmail.com (A.F.); rodrigo.sad.lima@gmail.com (R.L.); carlosresque@ufpa.br (C.G.S.); bianchi@ufpa.br (B.M.)
*   Correspondence: hortencio1983@gmail.com
†   These authors contributed equally to this work.

**Abstract:** The overlap of visual items in data visualization techniques is a known problem aggravated by data volume and available visual space issues. Several methods have been applied to mitigate occlusion in data visualizations, such as random jitter, transparency, layout reconfiguration, focus+context techniques, etc. This paper aims to present a comparative study of the reading of visual variables values with partial overlap. The study focuses on categorical data representations varying the percentage limits of partial overlap and the number of distinct values for each visual variable: hue, lightness, saturation, shape, text, orientation, and texture. A computational application generated random scenarios for a unique visual pattern target to perform location tasks. Each scenario involved presentation of the visual items in a grid layout with 160 elements (10 × 16), each visual variable had from three to five distinct values encoded, and the partial overlap percentages applied, represented by a gray square in the center of each grid element, were 0% (control), 50%, 60%, and 70%. Similar to the preliminary tests, the tests conducted in this study involved 48 participants organized into four groups, with 126 tasks per participant, and the application captured the response and time for each task performed. The results analysis indicated that the hue, lightness, and shape visual variables were robust to high percentages of occlusion and gradual increase in encoded visual values. The text visual variable showed promising results for accuracy, and the resolution time was a little higher than for the last visual variables mentioned. In contrast, the texture visual variable presented lower accuracy to high levels of occlusion and more different visual encoding values. Finally, the orientation and saturation visual variables exhibited the highest error and worst performance rates during the tests.

**Keywords:** information visualization; visual variables; evaluation; occlusion; overlap; visual perception

## 1. Introduction

The occlusion of visual items in 2D data visualization techniques is an old problem that has been observed in many studies presented in the literature [1–3]. The occlusion issue is generally exacerbated due to the visual space available to present the data and the amount of data to be displayed. Depending on the level of occlusion, the user´s perception of visual data can be affected, causing data misinterpretation or even loss of relevant information [4].

Considering previous studies, it is possible to highlight that they have mainly focused on the issue of reducing the occlusion effect for visual items when using data visualization techniques [5–7]. Some solutions reported include random jitter [8], the transparency technique [9], layout reconfiguration [10], focus+context techniques [11], multiple views [12], etc.

Studies within psychology indicate that humans have mechanisms for recognizing partially occluded objects even when reduced in size and with low resolution [13]. However, Rosenholtz [14] demonstrated that the recognition capacity of human beings decreases at higher levels of occlusion. Some research outside psychology has also suggested limits for occlusion levels that are tolerable for the human brain in efficiently detecting and

identifying partially overlapped visual elements [2]. In general, clues about the efficiency and effectiveness of partially occluded visual variables in information visualization are split across multiple articles, making it difficult to compare and use the correct visual variable for a given data visualization scenario.

Thus, this paper aims to present a comparative study of the perception of partially occluded visual variables in data visualization techniques, indicating how much the visual variable can be occluded and how many values it can encode in this condition. The results can be used as a set of good practices to apply visual variables in scenarios where occlusion is unavoidable,to increase the visual items in a data visualization technique, or even to provide indicators to compose quality criteria to evaluate data visualization techniques concerning the perception of their visual items. It should be noted that this study is an extension of research presented by Santos [15], in which the preliminary tests and initial results presented in the previous analysis are described in detail.

The comparative study proposal with regard to partially occluded visual variables is based on the location tasks for a visual target [16], and focuses on categorical data representations, considering hue, lightness, saturation, shape, text, orientation, and texture visual variables. To support the comparative study, a computational application was developed to generate random grid layout scenarios with 160 positions (10 × 16), where each grid element presents one encoded value (from 3 to 5 values) of one visual variable, including a gray square in the center of the grid element representing the level of partial overlap that should be applied to the visual item (0% (control), 50%, 60%, and 70%), with one visual target generated as the task aim. The test involved 48 participants organized into four groups of 12 participants with 126 tasks per participant; the application captured the response and time for each task performed.

The results analysis indicated that the hue, lightness, and shape visual variables maintained good accuracy and time performance for location tasks, even with 70% partial overlap and five values encoded. For the text and texture visual variables, the results showed good accuracy for partial overlap until 60% and until four values were encoded; the time performance of the text visual variable was a little better compared to the texture variable. For the 50% partial overlapping and three values encoded scenario only, the saturation variable showed good accuracy and time performance. Finally, the orientation variable was associated with the worst accuracy and performance results for all the partial overlap percentages and number of values encoded. However, the control group evidenced similar problems, which suggests the need for review of the visual encodes used.

This article is organized as follows: Section 2 presents the concepts relevant to the development of this study; Section 3 lists previous studies that address the perception of visual variables and provide analyses of the partial overlapping of elements; Section 4 presents the methodological decisions and protocols used in the configuration and execution of the tests in this research; Section 5 presents the results obtained and describes the collected data; Section 6 summarizes the main extracted results and highlights the recurring comments made by the participants; Section 7 discusses final considerations regarding this research and lists possibilities for future work.

## 2. Theoretical Foundation

In this section, the concepts related to the development of this study will be presented, such as the definitions and which visual variables were used, the evaluation approach used, and the statistical tests applied for analysis of the collected data.

### 2.1. Visual Variables

Visual variables can be defined as a set of visual characteristics that can be combined to create a unique visual item to convey information [17]. In the same study, the author proposed encoding visual information without ambiguity using seven visual variables (Cartesian plane [X, Y], size, value, texture, color, orientation, and shape) for four types of tasks (association, selection, ordering, and quantification).

As the present research focuses on improvement of the study presented in [15], the initial structures of the visual variables will be presented together with the modifications made and the final configurations of the visual variables analyzed.

### 2.1.1. Hue, Saturation, and Lightness

For this research, we used the HSL (hue, saturation, lightness) color system as a basis for the visual variables involving colors [18]. In this color system the hue's values vary between the pure spectrums of colors, starting with red, passing through green, and ending with blue. The saturation values range between gray and the value assumed by the hue. Finally, the lightness varies between black, passing through the color hue and ending with white [19].

The color spectrums described in [16,20] were applied in the study presented by [15] and were retained for this research. However, the HSL system was not used in the initial study, and the hue visual variable remained with the same characteristics. However, the saturation and lightness visual variables were not analyzed separately, which resulted in an analysis with discrepant results compared to the results presented in the following sections.

The spectrums used for the hue visual variable in [15] were the same as applied in this research. For the combination of the saturation and lightness variables, the purple color spectrum was used, which ranged from white to hue, thus not reflecting the HSL system.

### 2.1.2. Shape

Geometric shapes widely used in the consulted literature were used [16,17,21], including square, circle, cross, and star. However, since this study considered five (5) values per visual variable, a fifth visual coding value was added, a square shape with serrated edges.

In the previous study presented by [15], a diamond shape was used. However, it resulted in significant identification problems in the presence of other elements with smooth edge squares and circles, so the diamond shape was replaced by a serrated edge square shape, the structure of which stood out within the set of selected visual encodings.

### 2.1.3. Text

Visual coding values for the text visual variable are generally alphanumeric characters (letters and numbers). An example of the use of this variable is reported in the study presented by [22], in which the authors chose to use numbers to represent the continuous attribute values of the database.

Alphabet letters were used to represent the possible visual enconding values for the text visual variable assumed. The selection criterion was based on letters with visual structures that could differ as much as possible from each other. The text visual variable was used to represent categorical attribute values.

The letter "E" was initially added to the set of different values because it was present in [15]. However, it showed significant identification problems for all levels of partial overlap, so it was decided to replace it. The letter "W" was placed in its position. The set of all different values of the visual variables will be presented in the following sections of this paper.

### 2.1.4. Orientation

According to [17], the orientation visual variable has its structure formed by the change in alignment of the elements that compose it. Some studies reported in the literature [16,20,23,24] use a group of lines as the base element to form the orientation visual variable where their angles are modified to represent different values.

In this research, a group of arrows was used because the arrow's direction would provide additional information to identify different visual encoding values during the tasks and expand how the set of arrows could be arranged.

Compared to the study presented by [15], the different visual encoding values for the orientation visual variable were based on straight lines where the elements had their

inclination angles changed to represent these visual values, which resulted in many errors (accuracy) and difficulties in the identification tasks.

### 2.1.5. Texture

According to [17], texture is a visual variable whose structure is formed by variation in the granularity of the elements composing it. In this research, the texture structures were composed of circle shapes, and the different visual encoding values varied according to the increase in the number of circles that composed them.

Compared to the study presented by [15], the textures visual coding values composed of $3 \times 3$ and $5 \times 5$ circles were replaced by patterns with $4 \times 4$ and $6 \times 6$ circles. The replacement was due to the large number of errors observed in the preliminary study's identification tasks.

### 2.2. Evaluation Aspects

The evaluation was conceived considering certain key points, such as the tasks, participants, and data. The questions which guided the evaluation planning included, for example: What should be evaluated? In which order and how should the data be collected? What are the participants' profiles? and How should the participants be grouped? These questions and others will be detailed in the next sections.

### 2.2.1. Between-Subjects Design

A between-subjects design is a way of organizing groups that aims to avoid the effects that a participant can cause from their involvement in more than one group of analysis in the study [25]. The participants were divided into groups based on the partial overlap level independent variable. Thus, four groups (0%, 50%, 60%, and 70%) of participants were formed to perform the tests proposed, which will be described in more detail in the methodology section.

### 2.2.2. Within-Subject Design

A within-subject design is an organization in which all participants are exposed to all possible analysis scenarios [25]. In the case of this study, all participants in each analysis group performed tests involving all the evaluation scenarios proposed for tasks involving the analyzed visual variables.

An example of an evaluation scenario proposed for a task performed by one of the participants is "a participant from the 70% partial overlap group; six tasks involving the hue visual variable; with the hue visual variable encoding four different visual encoding values". It should be noted that these configurations will be presented in some more detail in the Methodology section.

### 2.2.3. Mixed Design

This study applied a mixed-design evaluation model, which seeks to perform analyses of visual variables between two or more different groups. Simultaneously, each group's participants are subjected to the same configurations of the proposed evaluation scenario tasks [25]. For this research, three independent variables were used: level of partial overlap, visual variables type, and the number of different visual encoding values. As mentioned earlier, each participant in the study performed tests involving each of the defined variables.

### 2.3. Bootstrap

Bootstrap is a statistical method Bradley Efron proposed using sampling techniques with replacement [26]. In statistics, a bootstrap is any random sampling test or metric with replacement. Use of the bootstrap method enables specification of accuracy measurements (based on measures of tendency, variance, confidence interval, or some other statistical measure) for estimates based on sampling techniques [27].

This method can also be used to develop hypothesis tests and is often used as an alternative to statistical inference based on the assumption of a parametric model when this assumption is doubtful, or where parametric inference is impossible or requires complex formulas to calculate the standard errors [27].

### 3. Related Work

For the development of this research, studies with the following characteristics were analyzed: tests of visual variable perception, object recognition with partial overlap, and applications or techniques that utilize partially overlapping visual objects. Studies that aimed to mitigate the impact of partial overlap were not selected.

Groop and Cole [28] evaluated two overlapping approaches based on the circle size. In the first approach, the overlapping circles were cut to fit together (sectioned), and, in the second, the transparency technique was applied to the overlapping circles. As a result, it was observed that the transparency proposal performed better. It is noteworthy that, in this study, the authors evaluated techniques for addressing the overlap problem, but did not consider different visual variables or overlap levels.

Cleveland [29] conducted perception tests in research using ten visual variables (common scale position, position unaligned scale, length, direction, angle, area, volume, curvature, shading, and saturation) to determine which visual variables best represented a set of quantitative data. In this study, the authors conducted an evaluation of visual perception of visual variables but did not account for the overlapping problem. The number of visual variables analyzed was a strength.

The authors in [21] presented another classification based on visual perception evaluations using 12 visual variables (position, length, angle, slope, area, volume, density, saturation, hue, texture, connection, containment, and shape). The classification was organized using three categories of data: quantitative, ordinal, and categorical. For the study proposal in this paper, we focused only on visual variables mapped as categorical data. In this study, the authors of the visual perception tests did not consider the overlapping problem.

Graham [30] described the challenges of visualizing structural changes in hierarchical techniques. The author suggested combining hue and shape visual variables to represent the leaf nodes in a tree diagram. Even with the proposed overlap treatment in mind, the study did not analyze multiple levels of overlap and utilized just a few types of visual variables.

In the research presented by [23], an analysis was conducted based on visual perception tests involving visual variables when applied to tasks of selection, association, ordering, etc. Bertin [17] was the source for the visual variables used. With regard to the findings presented in this article, a high level of visual perception was observed for the hue, texture, and orientation visual variables when used in selection tasks. Even though the authors of these studies did not conduct analyses regarding the overlapping problem, these studies are fundamental to the field of analysis of visual perception and visual variables.

Carpendale [23] conducted visual perception tests involving visual variables in tasks involving selection, association, ordering, etc. Bertin [17] was the source for the visual variables used. A high level of visual perception for the hue, texture, and orientation visual variables when used in selection tasks was highlighted.

Theron [31] proposed multidimensional glyphs to depict information with respect to a cinematographic dataset, where each component of the glyph represented a function performed by a member of a film's cast. The author created glyphs that were composed of varied overlapping elements (shapes), and each inserted element encoded a dataset attribute. The multidimensional glyphs effectively expanded the visual mapping of the additional base attributes, but the issue of overlapping glyph elements was not analyzed.

Brath [32] applied multidimensional glyphs in Venn and Euler diagrams. The author analyzed how combined hue and texture visual variables affected the user's visual perception. The results obtained indicated that the use of multidimensional glyphs for visual

mapping of multiple attributes of a dataset can be effective. Again, analyses on visual perception were reported this study, but without considering the issue of overlapping graphic elements.

In research conducted by Demiralp [33], hue, shape, and size visual variables were combined and evaluated in similarity and comparison tasks. The results indicated good performance for hue and shape visual variables used together. However, the authors did not consider analysis of the problem of graphic elements that overlapped.

In the study presented by Zhang [34], the authors presented a new visualization technique based on a focus+context approach, in which users were able to reduce the problem of overlapping elements. However, in this study, the authors did not perform visual perception tests directed at visual variables.

In a study by Soares [35], a multidimensional layer glyph concept was used, in which one visual variable overlays on another in some percentage, such that the visual variable in the top layer partially hides the visual variable in the lower layer. The author conducted visual perception tests for the glyph components. The data collected was analyzed by a decision tree algorithm, which generated good rules for building the multidimensional layer glyph, including the level of partial overlap and visual variables for each layer. The author's suggested glyphs were utilized in conjunction with the Treemap technique.

In his study, Pires [36] adopted the method proposed by Soares [35] for constructing multidimensional glyphs and designed a summarization glyph with a layered structure. Analyses were conducted involving employing the overlapping factor without compromising the legibility of the visual variables, which represents an important aspect of the study.

To validate his proposal for applying multidimensional glyphs in layers to squarified treemaps, Soares [37] presented an analysis based on visual perception tests on users, in which the visual variables used in the construction of the glyphs were influenced by the results of the research presented by Santos [15].

Korpi stated [2] that, despite evidence that humans can extract information from partially overlapping objects, no research has indicated which level of partial overlap a visual variable could have and still communicate information. Our study shares a similar motivation to this but goes further, increasing the number of visual variables analyzed and having different visual coding values, different levels of partial overlapping, and a context for the general application of visual variables independent of the mapped database.

The visual perception tests used three visual variables (color, shape, and pictograms) with equal dimensions for four partial overlap levels (0%, 25%, 50%, and 75%) in cartography scenarios (maps) [2]. The results indicated that the hue visual variable with 75% partial overlapping maintained good communication, and the shape visual variable with 50% partial overlap had similar performance compared to the pictogram visual variable without overlapping. Additionally, the author suggested that pictograms could be combined with a hue visual variable to improve their visual perception.

This study is an expansion of the research presented by Santos [15] and evaluates the robustness of visual variables to different visual encoding values (3, 4, and 5 different values) and different levels of partial overlap (0% (control group), 50%, 60%, and 70%).

Seven visual variables were selected from the literature: hue, lightness, saturation, shape, text, orientation, and texture. These visual variables were chosen due to their widespread use, evaluation, and application within the field of information visualization [2,17,21,30,33]. The numbers of different visual encoding values were defined based on the research presented by [23], in which the author indicated that these quantities can be effectively interpreted in scenarios without partial overlap.

The levels of partial overlap were defined based on Korpi [2] and Soares [35]. The first demonstrated that a visual variable could efficiently communicate information with 50% partial overlap. The second presented results showing that visual variable recognition decreases significantly with 80% partial overlap.

Importantly, in order to accomplish improvement of the initial results, changes were made to the analyzed visual variables and the visual coding values, as previously described

in the chapter entitled "Theoretical Foundation"; Figure 1 shows the initial visual variables analyzed and the visual coding values utilized in the preliminary tests.

| | Hue | | | Saturation | | | Texture | | | Orientation | | | Shape | | | Text | | |
|---|---|---|---|---|---|---|---|---|---|---|---|---|---|---|---|---|---|---|
| **DV** | 3 | 4 | 5 | 3 | 4 | 5 | 3 | 4 | 5 | 3 | 4 | 5 | 3 | 4 | 5 | 3 | 4 | 5 |
| **Subsets of Different Values** | | | | | | | | | | | | | | | | | | |
| **Configuration Default** | Hexadecimal Coding<br>#FF0101<br>#2C2CFF<br>#EBC089<br>#FFFF01<br>#41BA2F | | | Hexadecimal Coding<br>#EEC9E5<br>#D6A6C6<br>#BC86A9<br>#986995<br>#7C4D79 | | | Number of Circles<br>10 x 10<br>8 x 8<br>5 x 5<br>3 x 3<br>2 x 2 | | | Angle of Lines<br>180º<br>36º<br>72º<br>108º<br>144º | | | Geometric Shapes<br>Square<br>Circle<br>Lozenge<br>Star<br>Cross | | | Letters<br>A<br>C<br>E<br>K<br>J | | |

**DV - Different Values**

**Figure 1.** The initial visual variables analyzed in the study presented by Santos [15] are represented by their respective subsets of different visual coding values.

## 4. Methodology

In this section, all protocols followed for conducting the evaluation tests in this study will be described, such as the developed visualization scenarios, the computational environment used, the participants' profiles, and the applied evaluation scenarios.

### 4.1. Evaluation Procedure

The evaluation was conducted in a closed, climatized room with artificial lighting, where the participant was accompanied only by an evaluation facilitator. In addition, the distance between the monitor and the participants' eyes was approximately 70 cm, and the chair used by the participants had armrests and a backrest.

Before starting the test, each participant was invited to sign the FICF (free and informed consent form), where they were informed about the test goal, that the data collected during the tests would be used anonymously, and that they could withdraw from the tests at any time, regardless of the reason. They were asked to fill out a screening questionnaire with information about their age, gender, educational level, and restrictions in identifying colors. The study was approved by the Institutional Review Board (or Ethics Committee) of the 18-UFPA-Institute of Health Sciences of the Federal University of Pará (15522319.2.0000.0018 and 3 October 2019).

Since three of the analyzed visual variables refer to color (hue, lightness, and saturation), if the participant claimed to have difficulty identifying colors, the participant could not participate in the test.

After participants responded to the digital questionnaire, a training stage was provided to inform them of the research objective and to provide instructions on how the test would be applied. In the training stage, the participants were introduced to the application used in the test. At this point, six visualizations scenarios were presented to demonstrate to the participants how to respond to the tasks during the test. After the training, the test began.

After completing the tasks proposal, a questionnaire containing questions about the participants' perception of visual variables and their different visual encoding values was applied to collect data for analysis. The same sequence of stages was followed when conducting the initial tests [15] and when conducting the tests for this research. The flow of

the general procedure adopted for the evaluation carried out in this study can be seen in Figure 2.

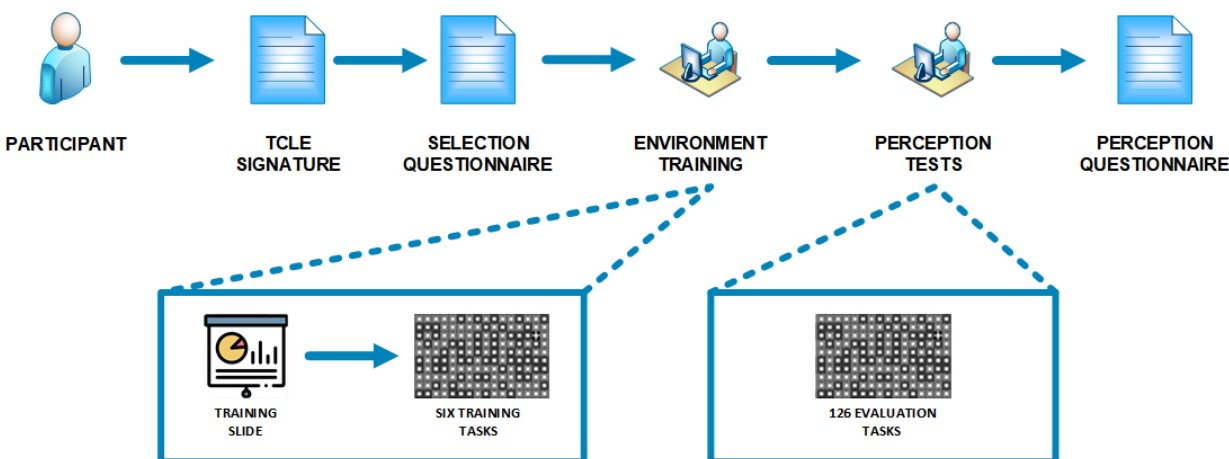

**Figure 2.** Each participant followed a sequence of steps in order to complete the evaluation tests.

### 4.2. Participants' Profiles

The tests conducted in this study involved 48 participants. All of them were from the academic community from the Federal University of Pará, aged between 18 and 42 years old, with education levels ranging from incomplete undergraduate to Ph.D., and belonging to various fields of knowledge, such as administration, biomedicine, natural sciences, computer science, etc. No participants declared having any difficulty related to color identification. No specific knowledge was required to participate in the study, such as familiarity with the concept of visual variables or information visualization techniques.

### 4.3. Computing Environment

A computer with 8GB of RAM, 1TB HD, and an Intel Core i7 processor was used to perform the tests. A 21″ monitor with a 1920 × 1080 pixels resolution was also used in landscape orientation.

The application developed for this study presented the visual variables in a 10 × 16 grid layout. The visual elements that composed each visualization were randomly generated, varying the type of visual variable, the percentage of partial occlusion, and the number of distinct encoded values. Examples of the visualization scenarios used to perform the evaluation tests can be seen in Figure 3.

Each evaluation task had a total of 160 elements containing only one distinct visual element (target item), varying the percentage of partial overlap level, the type of visual variable, and the number of different visual encoding values, as can be seen in Figure 3.

The visualization scenarios generated for the tests had a grid area size of 480 × 768 pixels. The size of each grid element was based on results presented by [35], where a decision tree algorithm applied to a visual variables perception dataset considering different area sizes indicated that the minimum area to perceive clearly visual variables would be 48 × 48 pixels. Figure 4 shows the visual variables and their respective visual values used in this study.

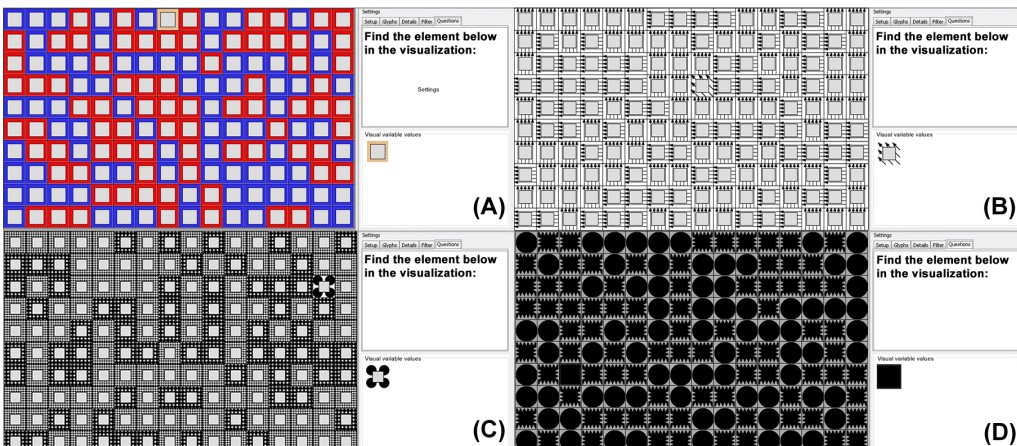

**Figure 3.** Examples of evaluation scenarios used in the tests: (**A**) Hue visual variable with 70% partial overlap, (**B**) orientation visual variable with 60% partial overlap, (**C**) texture visual variable with 50% partial overlap, and (**D**) shape visual variable with 0% partial overlap.

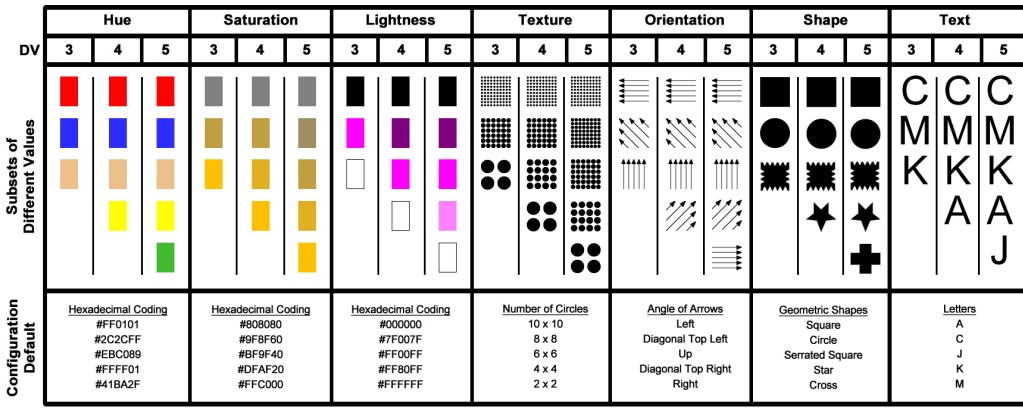

**Figure 4.** The visual variables analyzed in this study are represented by their respective subsets of visual coding values (3, 4, and 5 values).

### 4.4. Test Procedure

As general rules for conducting the tests, it was specified that participants should know the visual characteristics of the target element and be unaware of its location. For this, a visual element model required to be presented for searching in all the locate tasks performed by the participant.

This configuration was defined based on the taxonomy defined in [16], where different types of search tasks are specified. For this study, the type of search task defined was locate, which consists in having prior knowledge about visual characteristics of the target element but without knowing any information about its location.

Based on the preliminary tests and the initial results [15], it was possible to define some specifications regarding the composition of the visual variables and the time to complete each task, as described below:

- The visual coding values for each visual variable (Figure 4);
- Maximum time of 30 s to complete each task.

The test was designed as a mixed-design study [25] with three independent variables:

- Visual variables: hue, lightness, saturation, texture, orientation, shape, and text;
- Partial overlap levels: 0%, 50%, 60% and 70%;
- Number of different visual encoding values: 3, 4 and 5.

The participants were divided into four groups of 12 individuals, as suggested in [25], for a between-subjects design. The independent variables (visual variables and the number of distinct values) were permuted in 21 evaluation scenarios (visual variable × number of different visual encoding values × partial overlap level) and organized in a within-subject design [25]. For each evaluation scenario (21), the participant performed six (6) tasks, resulting in 126 tasks per participant. Figure 5 illustrates the distribution of the independent variables over the tasks.

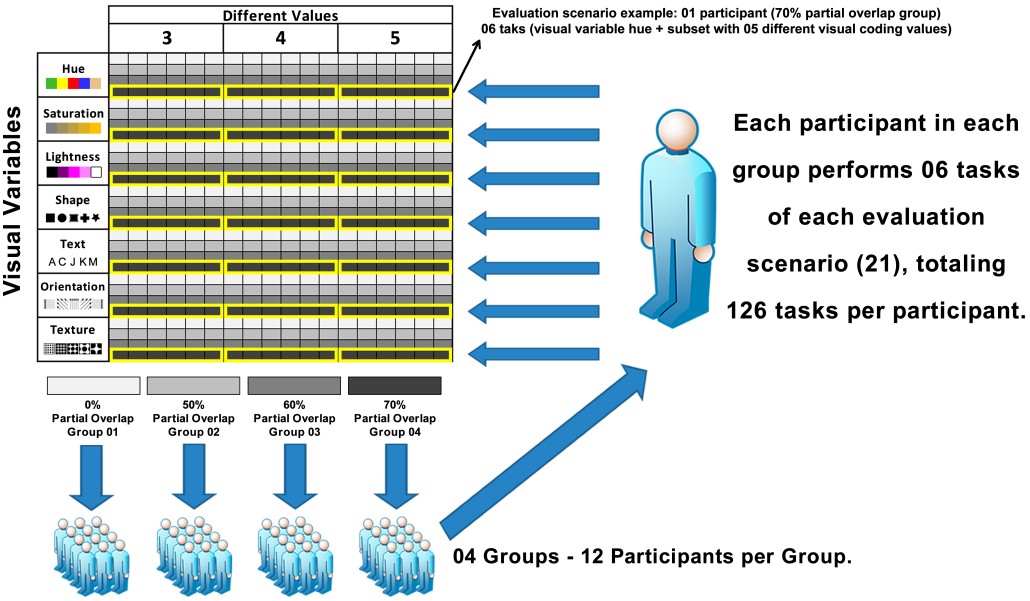

**Figure 5.** Organization of tasks by group of participants, considering visual variables, the number of distinct visual encodes, and the partial overlap level.

The following data were captured automatically for each test scenario:

- The level of partial overlap applied;
- The number of different visual encoding values per visual variable;
- The visual variable type;
- The target visual element and its characteristics;
- The participant answer: click on the correct item, click on the erroneous item, or no answer (time out);
- Task resolution time, which was quantified in seconds.

### 4.5. Statistical Analysis

The sampling with replacement bootstrap method was applied to the data collected from the tests. The bootstrap method was chosen because of its simplicity, the possibility to use the method for complex distribution with many strata (visual variables, different visual encoding values, and different partial overlap levels), and provision of accurate confidence intervals [38].

From the proposed test scenarios, two hypotheses were considered:

- Alternative hypothesis ($H_a$)—the performance analyzed from the overlap level (highest accuracy) and the resolution time (quickest resolution time) for the control group (0% occlusion) must return the best results;
- Null hypothesis ($H_0$)—that the application of occlusion would not affect the accuracy and resolution time.

The procedures followed by the sampling with replacement bootstrap algorithm, developed in the R programming language, are shown in Figure 6:

- The control group (0% occlusion) with 216 elements per visual variable was separated from the other groups (50%, 60%, and 70% occlusion) with 648 elements in total;

- The algorithm then drew 216 elements at random from each of the groups, always replacing the previously drawn element;
- After the simulation's final drawing round of 216 elements (for each group), the algorithm calculated the average of each group;
- If the control group value obtained was greater than that of the other occlusion levels group, then the alternative hypothesis would receive one point;
- According to [25], this procedure was repeated 20,000 times for each analyzed variable;
- $p$-Value calculation: $p = (1 − number\ of\ points)/20{,}000$;
- $p$-Value classifying: lack of significance ($p \geq 0.05$), moderate significance ($0.01 \leq p < 0.05$), strong significance ($0.001 \leq p < 0.01$), and extreme significance ($p < 0.001$).

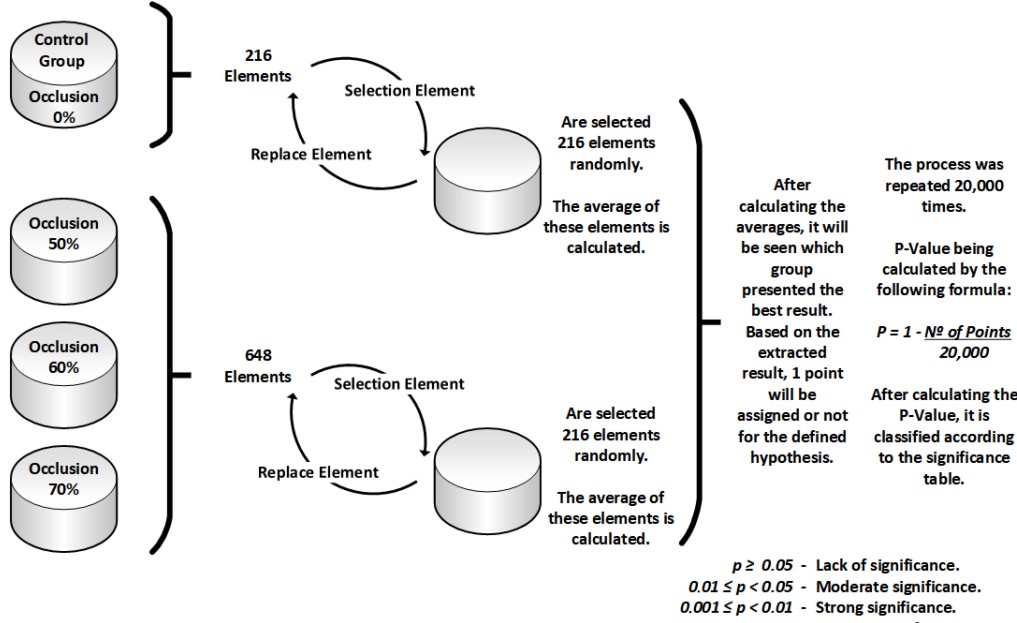

**Figure 6.** The hypothesis test procedure for estimating the effect of partial overlap level on the visual perception of each visual variable.

*4.6. Evaluation Scenarios*

The tasks' precision and resolution time scenarios were considered for results analysis based on the automatic logs. Moreover, a qualitative evaluation of each participant's perception of the visual variables partially overlapped was conducted.

For the accuracy analysis, three situations were considered: **TRUE**, when the element selected by the participants corresponded to the correct answer; **ERROR**, when the visual element chosen by the participant was an incorrect response; and **TIMEOUT**, when the 30 s timeout occurred. In addition, each type of answer was analyzed separately concerning the combinations of visual variables, the partial overlap levels, and the different visual encoding values.

Considering the partial overlap, a questionnaire was applied to collect the participants' opinions about their perceptions concerning the visual variables and their visual encodes. The participants were asked to classify the following items: the ease of recognition of the visual variables, and which visual encodes were poorly perceived.

The following scale was utilized to classify the performance of the visual variables qualitatively by participants:

- **GOOD**—All variables with a mean accuracy greater than 95%;
- **MEDIUM**—Visual variables with mean accuracy values between 90% and 95%;
- **POOR**—Visual variables with mean accuracy values of less than 90%.

In summary, the proposed analysis scenarios were:

- Accuracy per resolution time;
- Visual variables error rates;
- Classification of difficulty in identifying visual variables;
- Visual variables ranking based on participant perceptions;
- Analysis of visual encode values performance.

## 5. Results

This section presents the results obtained in two stages: initial and final. For the initial stage, the study focused on evaluating the performance of the different values selected for each of the studied visual variables [15]. The results allowed us to identify some problems in the sets of visual encodes chosen for the visual variables. In the next stage, the focus was on evaluating the visual variables under the partial overlap. It was proposed to gradually increase the level of partial overlap and the number of different visual encoding values to assess the perception of visual variables in each combined scenario. The two results stages that comprise this study are presented in more detail below.

### 5.1. Initial Results

In this section, the results presented by [15] highlight the issues identified in the sets of visual encode values of the analyzed visual variables (saturation, texture, orientation, shape, and text), and the proposed solutions to increase the efficiency of these visual variables to depict information on the partial overlap.

#### 5.1.1. Saturation

For the saturation visual variable, Santos [15] utilized a combination of the visual variables saturation+lightness. However, after the initial analyses, this combination was observed to show poor performance, resulting in an accuracy of less than 80% for the partial overlap scenarios and a total of 165 errors during execution of the tests.

Following this, the combination of saturation+lightness was separated into two distinct visual variables, and each was analyzed separately. The lightness variable accuracy increased by more than 98% with seven errors. The saturation visual variable accuracy decreased, reaching values of less than 80% with 193 errors.

#### 5.1.2. Texture

For the texture visual variable, Santos [15] used different quantities of circles to compose the texture pattern. The patterns were $2 \times 2$, $3 \times 3$, $5 \times 5$, $8 \times 8$, and $10 \times 10$ circles. From a total of 864 samples collected in the test, 80 were errors.

A significant error rate was observed when the texture visual variable encoded the $3 \times 3$ circles pattern (50%) or encoded more than three different visual encoding values. The final set of encoded values was modified for the following structures: $2 \times 2$, $4 \times 4$, $6 \times 6$, $8 \times 8$, and $10 \times 10$ circles. However, even after the changes, a new test found 87 errors. The difference from the last result was that there was no concentration on one specific visual encoding, and error increase was observed more in scenarios with five different visual coding values.

#### 5.1.3. Orientation

For the orientation visual variable, Santos [15] utilized a set of values for visual encoding based on basic lines and the variation in their respective angles. The angles used were 36 degrees, 72 degrees, 108 degrees, 144 degrees, and 180 degrees.

The orientation visual variable produced the worst result among all the visual variables analyzed, with an accuracy that remained consistently below 80% and with a total of 262 errors during the tests. After the initial results, the encoded values (basic lines) element was substituted by arrows, giving the participants an additional visual perception clue to compare the different visual values encoded. The changes decreased errors to 128, with an accuracy greater than 80% for all scenarios.

### 5.1.4. Shape

For the shape visual variable, Santos [15] obtained a preliminary result of 45 errors in 864 samples collected, around 5% of the total. Analyzing the errors, 90% of them were related to diamond visual encoding, with 36 errors. Following this, the diamond shape was replaced by a square with serrated edges. The shape visual coding value substitution reduced the error occurrences during the new tests, with a total of 12 errors out of 864 collected samples, with the visual coding value square with serrated edges not registering any errors.

### 5.1.5. Text

For the text visual variable, Santos [15] assigned the visual encode values A, C, E, J, and K. The initial results were 67 errors in 864 collected samples, with 45 errors referring to the visual coding value "E". The value "E" was replaced with "M" based on an analysis of the initial results, while the remaining values were maintained. The modification reduced the number of errors in the text visual variable to 41, representing a reduction of more than 40% of the initial results. The letter "M" only registered four errors during the tests.

### *5.2. Final Results*

New tests were conducted after modifying the visual encode set used by [15]. The analyses were performed following the evaluation scenarios described in the Methodology section, and the final results were presented.

### 5.2.1. Accuracy per Resolution Time

Initially, the correlation between the resolution time of the tasks, the mean accuracy of the visual variable types perception, and the level of partial overlap applied were analyzed for each group of combined values; Figure 7 shows the results of the data collection. The following presents some analysis of the scenarios:

- Mean accuracy levels of hue, lightness, text, and shape visual variables were classified as GOOD and MEDIUM (consistently over or equal to 90%);
- The hue visual variable had the best resolution time (constantly less than 2.5 s);
- The partial overlap effect on the mean accuracy of the saturation visual variable when mapping three different values was minimal;
- The lightness and shape visual variables showed very similar performances, which had to be distinguished by resolution time analysis, which demonstrated that lightness had the superior performance;
- The number of encoded values had the most significant impact on the resolution time of the orientation visual variable (consistently over ten seconds);
- The progressive increase in the number of different visual encoding values significantly impacted the mean accuracy of the saturation visual variable in all of the proposed scenarios (mean accuracy ranged from 74% to 82%);
- The visual variable most affected by the progressive increase in partial overlap level was the text visual variable, whose mean accuracy presented values around 90% (the 70% partial overlap scenario).

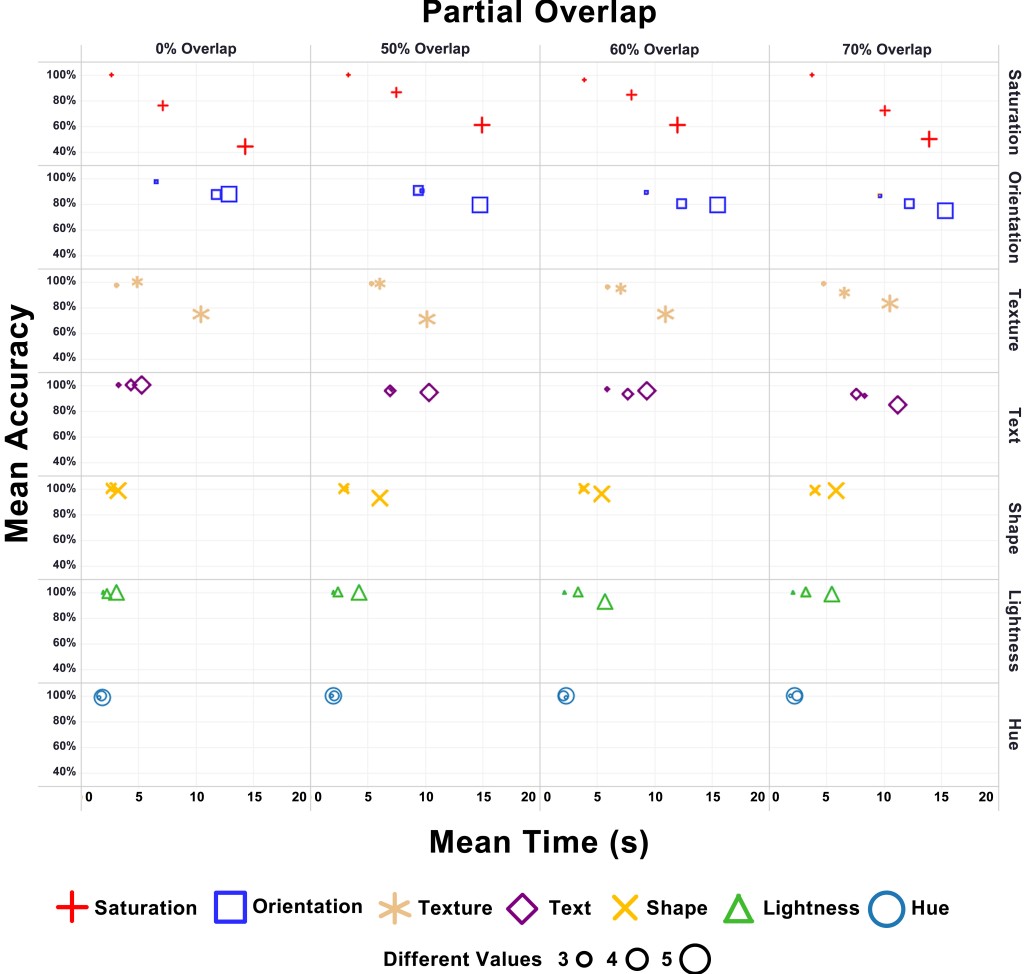

**Figure 7.** Resolution time, mean accuracy, and partial overlap for each proposed scenario of visual variables studied, partial overlap levels, and the number of different visual coding values.

Statistical tests based on hypotheses regarding the correlation between the accuracy and the resolution time were applied to each studied visual variable to determine the degree to which partial overlap influenced each.

Figure 8 (Accuracy) and Figure 9 (Resolution Time) illustrate the influence of partial overlaps on each visual variable, as determined by the selection and comparison of twenty thousand samples (bootstrapping method).

Figure 8 shows that the text visual variable was most affected by the partial overlap (*p*-value = 0 and a 6.33% reduction in its accuracy), followed by the orientation visual variable (*p*-value = 0.013 and a 7.41% reduction in its accuracy). The other visual variables did not reach a sufficient statistically significant difference to draw a conclusion.

Considering the resolution time of tasks, Figure 9 shows that the text visual variable was the most affected by the partial overlap (*p*-value = 0 and a rise of 2.54 s), followed by the shape visual variable (*p*-value = 0 and an increase of 1.12 s), the lightness visual variable (*p*-value = 0 and a rise of 0.87 s), and the hue visual variable (*p*-value = 0.0005 and an increase of 0.31 s). The other variables did not show statistically significant differences.

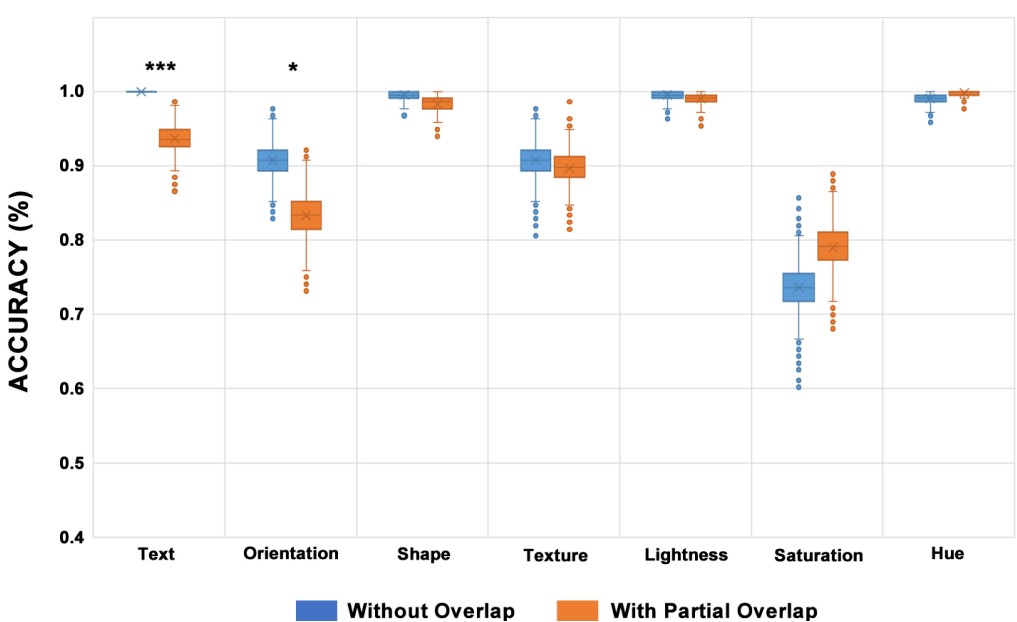

**Figure 8.** Bootstrap results with and without partial overlap (*n* = 216). Text (*p* < 0.001 ***) and orientation (*p* < 0.05 *) visual variables showed a difference in significance at alpha = 0.05 due to partial overlap.

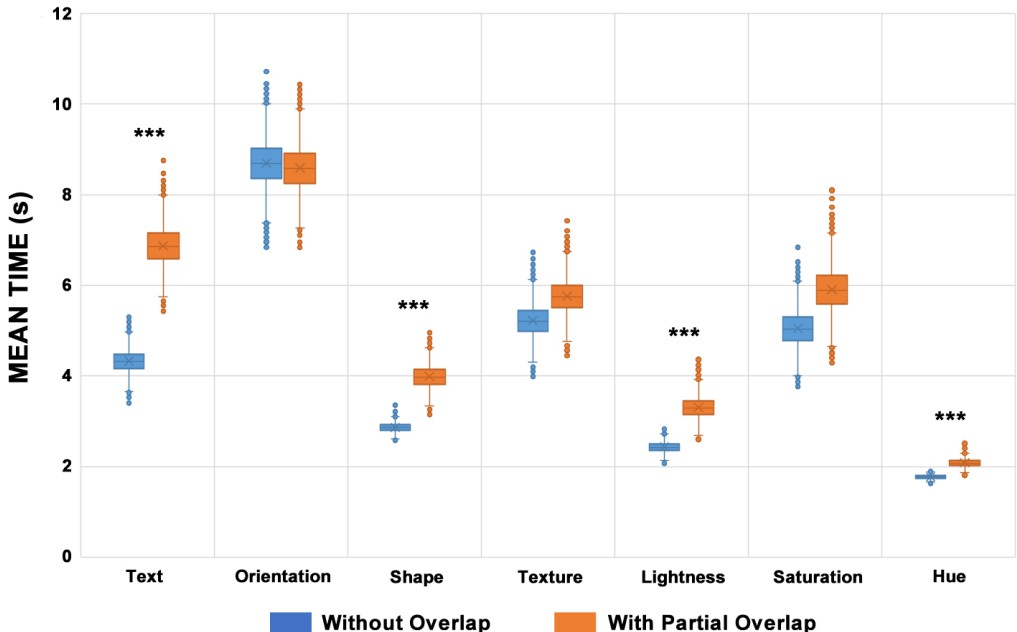

**Figure 9.** The mean time of the bootstrap results with and without partial overlap (*n* = 216). Text, shape, lightness, and hue visual variables showed a statistically significant difference because of the use of partial overlap (*p* < 0.001 ***).

### 5.2.2. Error Rates of Visual Variables

The number of errors (incorrect clicks) and timeouts (when the participant did not answer within the task time limit) were analyzed. Figure 10 shows the number of errors for each visual variable, considering ERROR (blue) and TIMEOUT (orange).

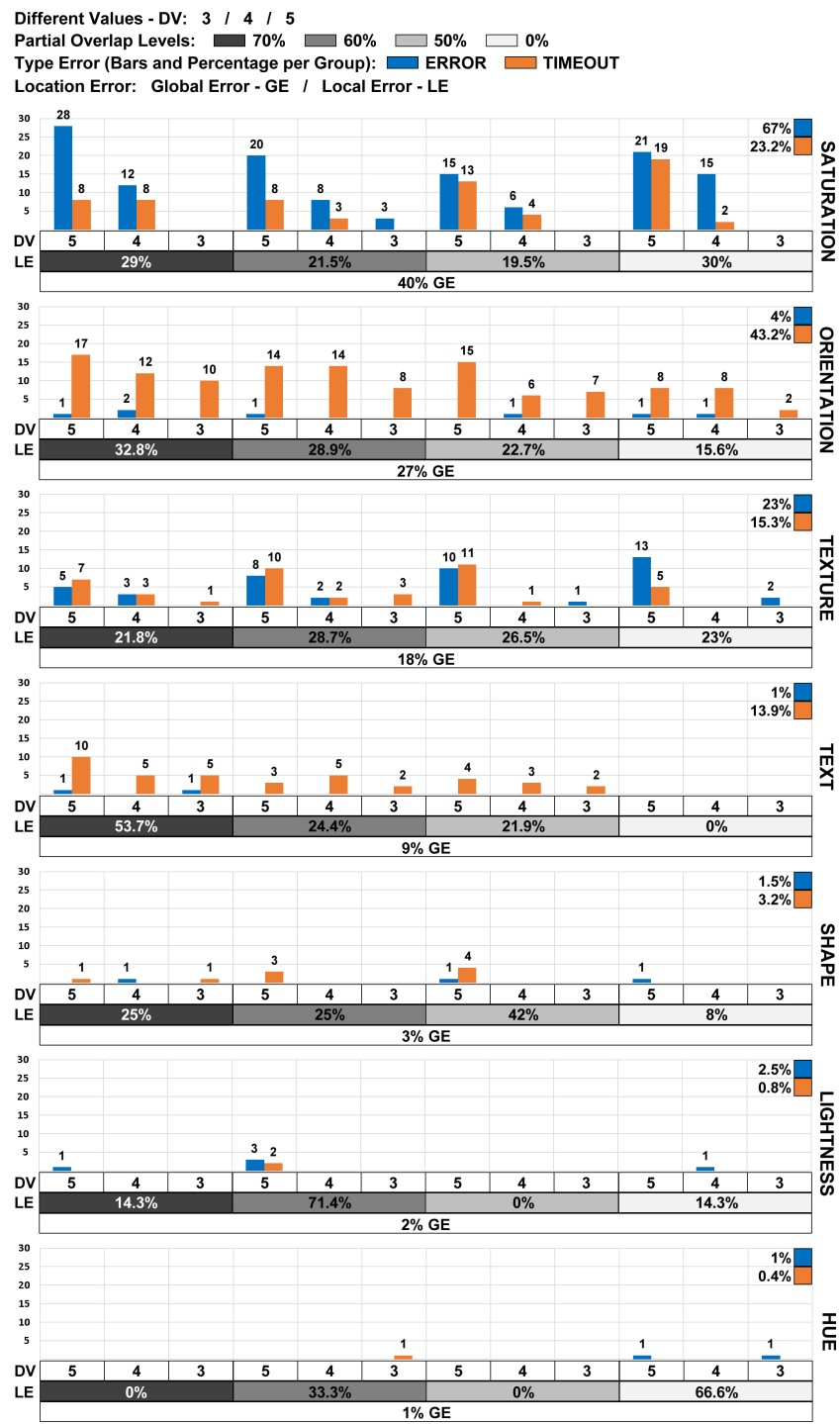

**Figure 10.** Analysis based on the number of ERRORS and TIMEOUTS for each visual variable. The saturation visual variable showed 40% of all errors registered considering all proposed scenarios.

The saturation and orientation visual variables had the highest number of ERRORS and TIMEOUT. For ERRORS, the saturation and orientation visual variables showed 40% and 27% of all registered occurrences, respectively. Specifically, the saturation visual variable with the three encoded values scenario presented fewer errors. Moreover, the saturation and orientation variables obtained the highest TIMEOUT occurrences with 23.2% and 43.2%, respectively.

The results obtained for the saturation visual variable could indicate similar encoded values, confusing the participant in scenarios with more than three different visual coding values.

The accentuated number of TIMEOUT occurrences for the orientation visual variable suggests the necessity to review the entire visual encode proposed for the visual variable. Even for the control group, there were excessive TIMEOUTS, mainly for scenarios with a higher level of partial overlap and more than three different visual values.

The texture and text visual variables presented 18% and 9% of all ERRORS registered during the tests. For TIMEOUT, the visual variables presented 15.4% and 13.3% of the total occurrences, respectively. In this context, it is essential to highlight the difficulty observed in identifying the "K" visual coding value (55% of the total events of TIMEOUT for this visual variable), indicating that this specific visual coding value must be replaced with another one.

The shape, lightness, and hue visual variables had the lowest numbers of ERRORS during the tests with, respectively, 3%, 1% and 1%. For TIMEOUT, the visual variables presented 3.2%, 0.7% and 0.3% of the total occurrences, respectively. In general, the results support the conclusion that these visual variables are robust to high percentages of partial overlap (70%), even for five different visual encoding values per visual variable. It was possible to identify several errors with the CROSS visual encode, when it corresponded to the target item. The lightness visual variable also showed some ERRORS in a specific scenario with five different values.

Figure 11 shows the total number of errors (ERRORS+TIMEOUT) for each evaluation scenario (visual variable + overlapping level + subset of different visual encoding values). The scenarios with saturation or orientation and five or four different visual coding values exhibited worse results in accuracy and time.

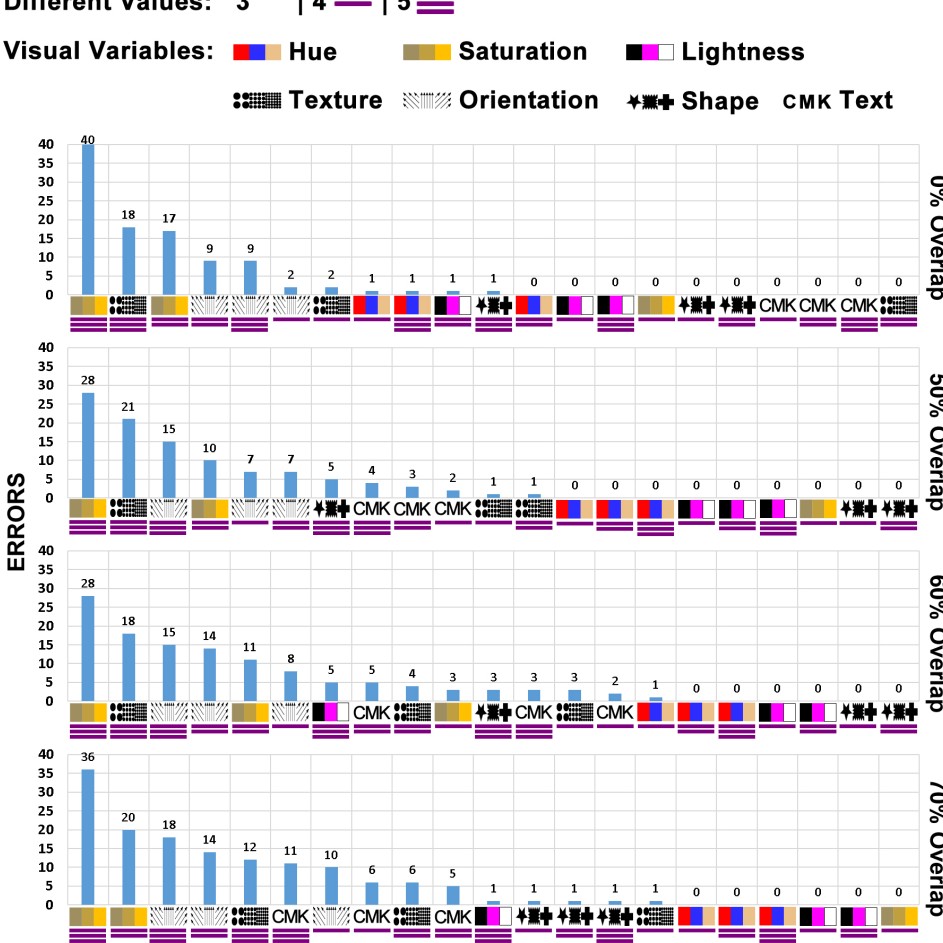

**Figure 11.** Total number of errors (ERRORS + TIMEOUT) for each evaluation scenario (visual variable + overlapping level + subset of different visual coding values).

The data were ordered from the highest number of errors to the smallest, with 40 errors being the maximum number of errors in one evaluation scenario. During the data analysis, the saturation, orientation, and texture visual variables had the most significant number of errors.

### 5.2.3. Visual Variables Ranking Based on Participants' Perceptions

The participants ranked the visual variables according to the facility level to identify the target item during the tests. Figure 12 shows a heatmap representing the ranking obtained from participants' answers, using a divergent color scale from easiest (green) to most difficult (red).

The visual variables ranking to identify the target item more easily was hue, shape, lightness, text, texture, orientation, and saturation. The shape visual variable was considered to require less effort than the lightness visual variable when identifying the target item in the tasks. However, they were practically tied when observing the difficulty level for both variables.

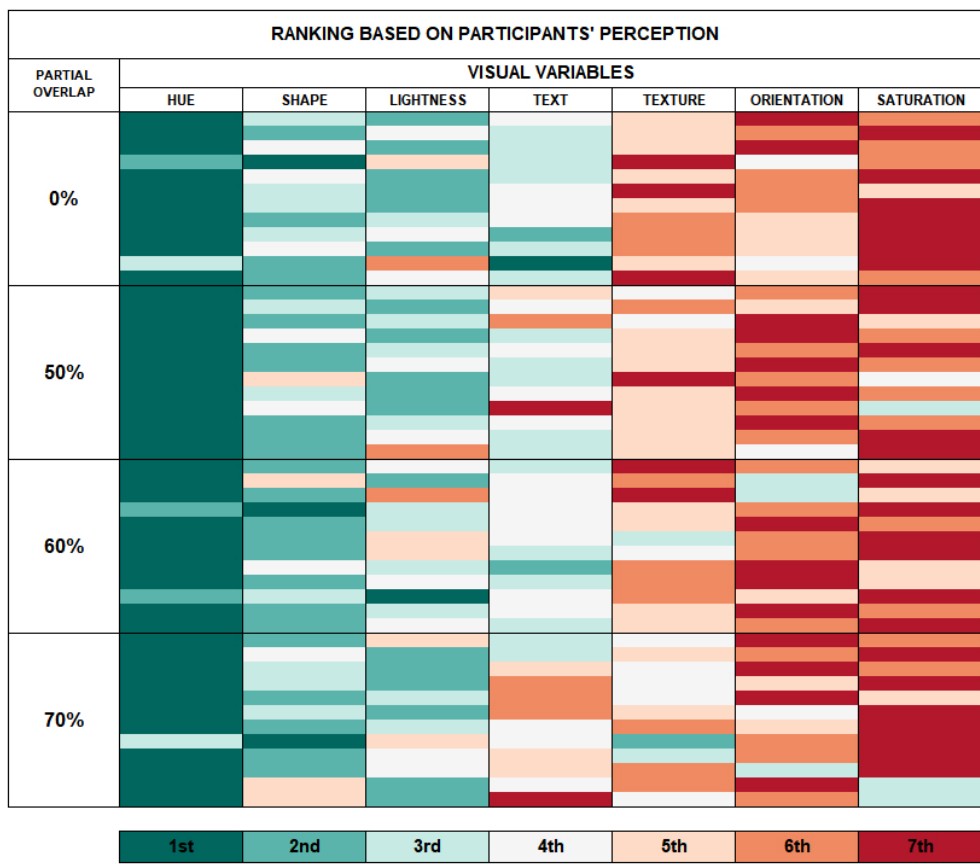

**Figure 12.** The ranking of visual variables to identify the target items more easily. The ranking presented from the visual variable considered easier to the visual variable considered more difficult to identify.

### 5.2.4. Performance Analysis of Visual Coding Values

An error ranking of visual values was undertaken from data obtained automatically from the application developed for the evaluation test. This ranking was compared with the difficulty ranking for each visual value to identify target items in the proposed tasks, and the difference in the positions in both rankings was calculated. In total, 471 data errors were collected automatically, while 478 answers for the identification problems were obtained.

The primary purpose of the comparison was to find visual values with low and high discrepancy positions considering both rankings. The comparison results could indicate true correlations (if the visual item is good or bad for the identification task), false positives,

and false negatives. From the 35 different values of visual coding used, those with more than ten errors were considered to have a significant identification problem.

Figure 13 presents the two rankings, highlighting the values with the most significant differences between the number of errors registered using the testing tool and the number of poor perceptions registered by participants through the applied questionnaires.

**Test Tool**

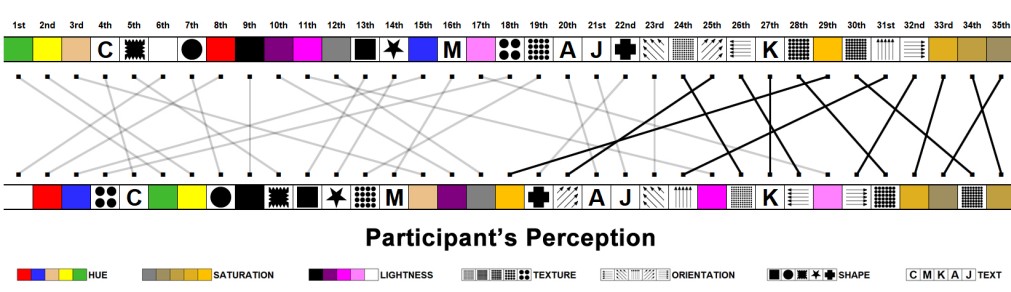

**Figure 13.** Visual encoded items with the most significant position difference in both rankings. Highlighting the elements in the 29th (saturation) and the 31st (orientation) positions.

The first analyses showed that the hue, lightness, saturation, texture, and orientation visual variables presented the highest levels of position variance. The only values with more than ten recordings of errors were the 29th (saturation-test tool 34 records and perception questionnaires 8 records) and 31st (orientation-test tool 39 records and perception questionnaires 19 records). Additionally, some visual items were reported as worse perception by the participants, contrary to the fact that the test tool did not register significant errors.

A second analysis was undertaken to determine when the visual coding values that recorded more than ten errors from the test tool indicated a direct relationship with the participants' perception. Figure 14 shows both rankings, emphasizing the visual coding values that recorded more than ten occurrences of errors by the test tool.

**Test Tool**

**Figure 14.** The visual coding values with the highest number of errors obtained by the test tool compared with the number of identification difficulty perceptions registered by the participants.

The results of the second analysis demonstrated a more effective relationship between the objective and the subjective data. The analysis enabled the determination of the visual coding values that presented the highest identification problems. At this point, it should be emphasized that many of the visual coding values of the saturation, orientation, and texture visual variables had elevated identification problems.

A visual coding value that diverged from the standards and stood out in this analysis was "K" (text visual variable), which presented a high number of testing tool errors (23 errors) and worse perception indications (20 indications).

## 6. Discussion

This section presents a summary of the study's findings for each of the visual variables analyzed, focusing primarily on the quantitative (evaluation tasks) and the qualitative (user perception questionnaires) metrics used.

### 6.1. Hue

For all the proposed analysis scenarios involving partially overlapping levels and subsets of different visual encoding values, the hue visual variable proved to be robust. It was the visual variable with the lowest number of identification errors (testing tool) and participant observations (questionnaires) regarding identification difficulty.

Following the analyses conducted in this study, the hue visual variable was ranked as the most robust visual variable reflecting a gradual increase in the partial overlap level and the number of different visual encoding values. Its mean accuracy was between 99.6% and 100% (GOOD).

### 6.2. Lightness

The lightness visual variable showed a high robustness level, close to that of the hue visual variable accuracy result. Its mean accuracy reached 99% (GOOD), and it did not show increased identification problems in any evaluation scenario proposed. A total of seven errors were recorded during the evaluation tests, which represented two percent of all the global data collection errors. Regarding the overall performance, the lightness variable ranked second among the studied visual variables.

The difference between the results extracted from the data collected via the test tool and the applied questionnaires is emphasized. More records (57 records) from participants were observed than for the data collected via the test tool (seven errors). Even though participants reported a small degree of difficulty in identifying the visual coding values, the lightness visual variable presented a high level of accuracy for mapping categorical data in all the evaluation scenarios considered in this study.

### 6.3. Shape

The shape visual variable showed a high level of robustness for all the percentage levels of partial overlap, with a minor increase in the resolution time compared to the scenario without partial overlap (control group). However, most errors with the shape visual variable occurred in scenarios where the cross visual value represented the visual target, which was also reflected in the analysis of the participants' opinions.

Eleven participants registered difficulty in locating the cross visual coding value when it was adjacent to elements encoding the square value. Nevertheless, the shape visual variable exhibited 12 errors during the evaluation tests, representing just three percent of the global errors. Finally, the shape visual variable ranked third among the visual variables analyzed, with a mean accuracy ranging from 98% and 99% (GOOD).

### 6.4. Text

When analyzing the scenario without partial overlap, the text visual variable had the highest accuracy (100%) among the visual variables studied. It obtained good results, with a slightly negative effect on the mean accuracy and the resolution time as the level of partial overlap increased; its mean accuracy varied between 90% and 100% (varying between MEDIUM and GOOD). In terms of performance, it ranked fourth among the studied visual variables.

Most participants in groups with overlapping levels could not easily locate the "K" visual item. This identification difficulty can be explained by the fact that the letter "K" has a rectangular shape similar to the overlapping element used in this study (gray square). Participants noted that the visible portions of the "K" visual item were more challenging to locate than those of the other letters.

This analysis may suggest that the text visual variable when overlapped with a polygon approximating the shape of the encoded value may impair the participant's visual perception; however, additional research with different overlap shapes is required to reach more conclusive conclusions.

It is worth highlighting that the text visual variable was the one that diverged most from the results found in the literature consulted [22] as its performance in scenarios with partial overlap levels was better than expected.

### 6.5. Texture

The accuracy and resolution time results of the texture visual variable ranged from POOR to MEDIUM (88% to 91%), placing it fifth in terms of performance among the visual variables studied.

From the questionnaires, it can be seen that the texture visual variable was associated with the most divergent opinions. The participants identified the visual items $6 \times 6$, $8 \times 8$, and $10 \times 10$ circles as having a more difficult identification level. It is also important to note that there were no comments regarding coding of the $2 \times 2$ circles.

It was found that the $2 \times 2$ and $4 \times 4$ circle encodings were the easiest to identify. These results suggest that fewer texture encodings were simpler to distinguish, which is supported by the low error rate when the visual variable maps only three distinct values.

Even so, good performance was obtained for the subset in which four different visual encoding values were used for the texture visual variable ($2 \times 2$, $4 \times 4$, $6 \times 6$, and $10 \times 10$ circles) as the number of test-related errors was reduced.

Lastly, the results suggest that the texture variable for the encoded values was not a robust visual variable when considering the gradual increase in the number of different visual encoding values and the percentage of partial overlaps.

### 6.6. Orientation

Based on the participants' comments, the orientation visual variable presented a high level of difficulty in identifying visual targets. From the results obtained, the following values for the accuracy of the orientation visual variable were obtained: 81% for 70% partial overlap (POOR classification), 83% for 60% partial overlap, 87% for 50% partial overlap (POOR classification), and 91% without any occlusion (MEDIUM classification).

The orientation visual variable also received several negative comments and had significant errors (128 errors using the test tool) for all the visual coding values. At the end of the analyses, this variable ranked sixth among the studied variables.

The mean time for solving tasks involving this visual variable was considered high, above 10 s. All the visual values received many negative comments, suggesting that the variable had general identification problems for scenarios with partial overlap.

Finally, it should be noted that the values based on the angles of 45° and 135° showed the lowest number of errors and reported negative comments from the participants.

### 6.7. Saturation

Even though the saturation visual variable demonstrated good robustness in scenarios of three distinct values, it presented heightened levels of difficulty in scenarios for four or five different visual encoding values.

The accuracy of the saturation visual variable ranged between 74% and 82% (POOR), placing it in seventh and last place in the performance classification of the visual variables studied. It is worth highlighting that the problem with this visual variable occurred at all percentages of partial overlap and resulted in many identification errors (test tool—193 in total). The saturation visual variable received many negative comments from the participants (questionnaire—137 in total).

When considering the gradual increase in the number of different visual encoding values, the saturation visual variable also demonstrated accuracy issues. This can be seen in the analysis of the results obtained from the control group (without partial overlap), in which the accuracy obtained was only 44% when five different values were mapped; this percentage is considered low for analysis in a scenario without partial overlap.

## 7. Final Remarks and Future Work

The objective of this research was to evaluate the visual perception of hue, saturation, lightness, texture, orientation, shape, and text visual variables when used for the visual mapping of categorical data and gradually increasing the level of partial overlap (0%, 50%, 60%, and 70%) and different visual encoding values (3, 4, and 5) in the context of location search tasks [16] with a single-value target item.

In the evaluation tests, 48 participants were divided into four groups of 12, one for each level of partial overlap [25]. The results obtained demonstrate that the primary objective of this study, the formulation of a classification for visual mapping of categorical data, was achieved.

The classification resulting from this research could help to direct future studies or may provide opportunities for application when visual variables are analyzed by professionals in the fields of information visualization, data science, and the development of data-driven interfaces. The recommendations presented here are based on the findings of this study.

Analyzing the classification obtained in this study, the hue visual variable, which produced the best results for both quantitative and qualitative data, was found to be the best visual variable analyzed for the visual mapping of categorical data.

The lightness and shape visual variables performed well according to the evaluation metrics employed in this study, and, despite the presence of some negative comments from participants via the qualitative questionnaires, they were classified as good visual variables for performing the visual mapping of categorical data with a high level of partial overlap and number of different visual encoding values. These variables were placed as the second and third top visual variables that were analyzed.

The text visual variable showed positive findings for the control group (0% occlusion); however, it was weaker when the level of partial overlap was gradually increased. For this visual variable, care should be taken when mapping categorical data, evaluating the visualization technique to be used, and determining whether there is a possibility of graphic elements overlapping.

Due to the high number of errors occurring at all levels of partial overlap and variation in the number of different visual encoding values, the texture visual variable was not classified as a good option for the visual mapping of categorical data, suggesting that, if the user chooses to use it, several factors must be evaluated (e.g., occurrence of partial overlap of elements, number of visual coding values, etc.).

The orientation visual variable was found to be unsuitable for performing the visual mapping of categorical data because it yielded unsatisfactory results across all the evaluation scenarios considered in this study (all levels of partial overlap and number of different visual encoding values).

Even though the subset of three different visual encoding values yielded good results for all levels of partial overlap, the performance of the visual variable saturation was significantly impacted by a gradual increase in the number of different visual encoding values (four and five). Therefore, it is suggested that this visual variable be used to map categorical data in scenarios where mapping a small number of different visual encoding values (three or less) is required.

As a summary of the findings and the correlations for the visual "variables" × "partial overlap" × "different visual encoding values", rankings of the groups are shown in Figure 15.

The following possibilities for extension of the research are suggested:

- Evaluation of other types of search tasks, such as direct search, navigation, and exploration [39], as well as similarity, comparison, and grouping;
- Evaluation of the partial overlap at other levels, thus expanding the results presented here;
- Evaluation of additional visual variables, including transparency, arrangement, lightness, focus, and resolution;
- Consideration of partial overlap when proposing novel InfoVis designs (e.g., treemaps + multidimensional glyphs).

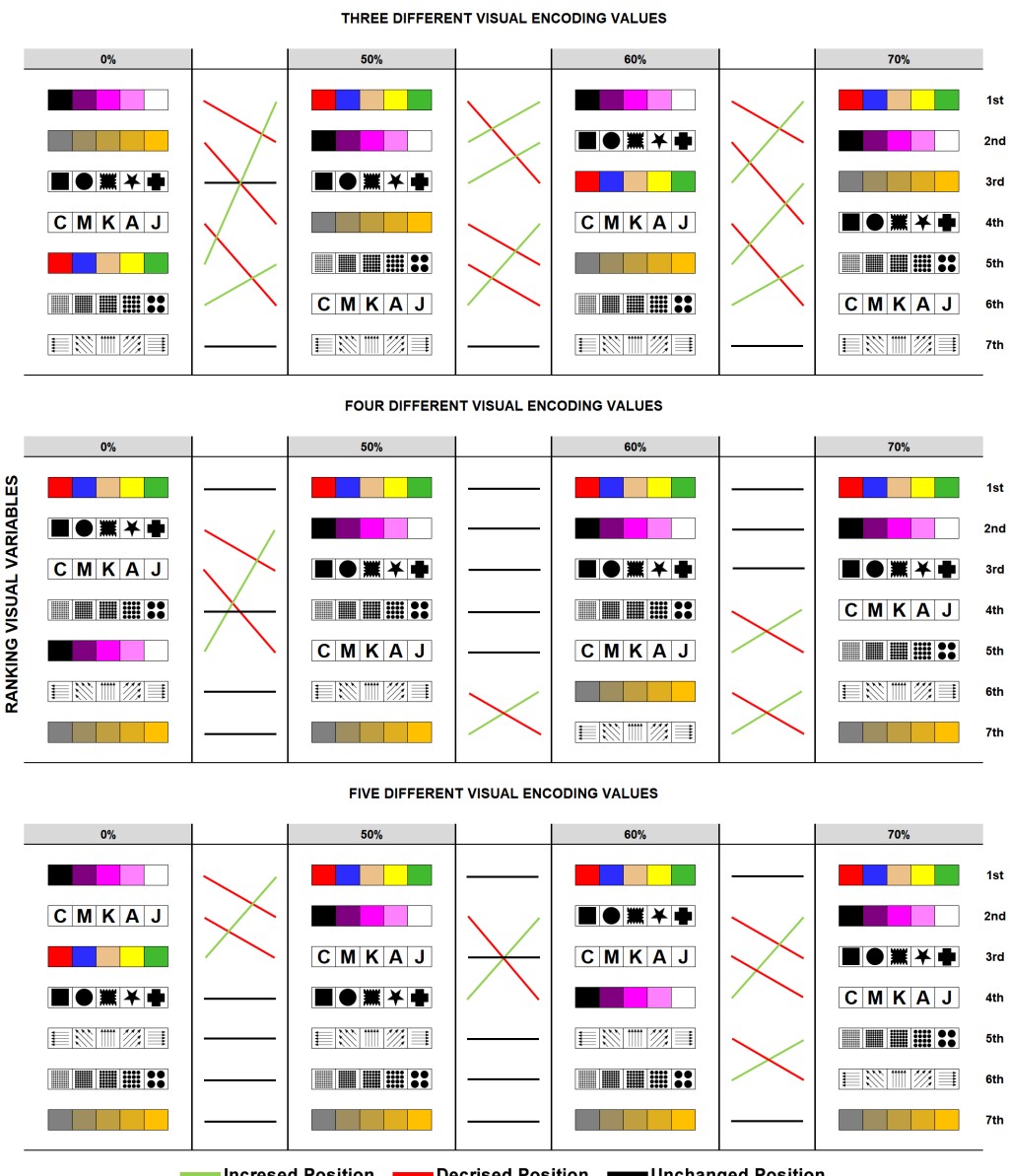

**Figure 15.** Ranking of the visual variables for different percentages of partial overlap and different quantities of visual coding values.

**Author Contributions:** Conceptualization, D.S. and B.M.; Methodology, D.S. and B.M.; Software, D.S. and C.G.S.; Validation, D.S., C.G.S. and B.M.; Formal analysis, D.S., R.L. and C.G.S.; Investigation, D.S. and A.F.; Resources, B.M.; Data curation, D.S. and R.L.; Writing—original draft, D.S., B.M., C.G.S. and B.M.; Writing—review & editing, D.S., A.F., R.L. and B.M.; Visualization, D.S., A.F., R.L. and B.M.; Supervision, B.M.; Project administration, C.G.S. and B.M. All authors have read and agreed to the published version of the manuscript.

**Funding:** This research APC was funded by the Federal University of Pará (UFPA).

**Institutional Review Board Statement:** The study was approved by the Institutional Review Board (or Ethics Committee) of 18-UFPA—Institute of Health Sciences of the Federal University of Pará (15522319.2.0000.0018 and 3 October 2019).

**Informed Consent Statement:** Informed consent was obtained from all subjects involved in the study.

**Acknowledgments:** The authors thank the Federal University of Pará (UFPA), the Graduate Program in Computer Science, the professors and students of the research from the Laboratory of Visualization,

Interaction and Intelligent Systems (LabVIS), and to Anderson Gregório Marques Soares from the Federal Rural University of the Amazon.

**Conflicts of Interest:** The authors declare no conflict of interest.

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
