# Peer review of "Investigating the Impact of Different Partial Overlap Levels on the Perception of Visual Variables for Categorical Data"

_applsci, doi:10.3390/app13169268_

Round 1

Reviewer 1 Report

The authors have submitted this manuscript on a pre-print server which is why it shows full of similarity in the abstract and introduction section.

Most importantly, the figures tables, and equations should be cited in the text & the insertion of the same should be very near to where it is cited. Further, when writing abbreviations, must write the full names when these first appear on the paper. Further, the full form of abbreviations must be used in the manuscript.

I recommend including a table of the previous study in the related work section very precisely with their proposed solution. The authors can take help from the following sample paper and can cite this paper for a better understanding of their method.

https://doi.org/10.1155/2020/8833767

What is the pilot test?

What type of different visual encoding values are used and how?

Regarding the conclusion paragraph, Please precisely describe the outcome of the study and justify the statements that are mentioned in the abstract. Further, it must contain additional points and must give a clear and more discussion about the experimental results. The main novelty and contribution of needs must be summarized and highlight the recommendations based on obtained results. These results are the hallmark for future extension therefore, please spend some more time writing the conclusion and based on the results suggest new directions.

While reviewing the references, I observed that cited references such as [12], [17], and [23] are outdated and not relevant whereas more work already has been done in the proposed study. The cited references are neither sufficient nor suitable and therefore must extend the list and focus only on the papers from the recent 3 years. The following references may be added to supersede the outdated ones for the authors' convenience.

https://doi.org/10.46565/jreas.2020.v05i02.006

Go for a thorough proofread of the paper to rectify several existing typos and grammatical mistakes to improve the written quality of the paper. If necessary take the help of a native English speaker to improve the language of the paper.

Author Response

Thanks for your suggestions.

The corrections made are shown in the attached file.

Reviewer 2 Report

In this study, the authors aimed to provide a comparative analysis on reading visual variables values with partial overlap. The focus of this study was on categorical data representations varying the percentage limits of partial overlap and the number of distinct values for each visual variable.

The findings are very relevant to current data visualisation approaches. This represents novel addition to existing literature.

Full proficiency and able to convey the message accordingly.

Author Response

(The authors gave the same response as above.)

Reviewer 3 Report

The following are the suggestions towards improvements in the paper:

1. The related work section can give glimpses on gaps identified and hence novelty or contribution of the work as well as objectives.

2. The methodology section could be detailed in terms of work flow diagram or block diagram.

3. some observations and inferences can be drawn based on the results

4. The inferences could lead to concrete conclusions.

5. Include more recent references and cite them as part of literature survey.

Author Response

(The authors gave the same response as above.)

Reviewer 4 Report

This paper presents a comparative study to read visual variables values with partial overlap in data visualization techniques. The obtained results can be applied in scenarios where occlusion is unavoidable to increase the visual items in data visualization technique. The reliability of the experience results is verified by experiments. There still exists the following issues:

1. Page1, lines1-19. Abstract cannot clearly reflect the practicality of this article’ findings, and it is necessary to emphasize where the findings of this article can be applied in specific areas.

2. Page1, lines22-28. Can you provide a specific figure in the introduction to address the occlusion problem of visual items? so that readers can have a more intuitive understanding.

3. Page11, lines 399-401. Why set the mean accuracy of variables corresponding to GOOD, MEDIUM, and POOR as greater than 95%, between 90% and 95%, and less than 90%? Please explain in detail.

4. Is the title of Figure 8 incorrect? Perhaps it should be changed to ‘Resolution Time’. Because I saw that you were using this word on line 493.

5. Page17, line549-553. Why is the color of the corresponding easy scale in Figure 17 blue, but what do you mean here is green?

6. Some sentences in this article are too cumbersome and verbose. The language should be simplified and adjusted.

7. The format of figures in the paper is not standardized. For example: the title in Figure 4 is not centered; The headers in almost all figures should be the consistent size; Figure 9 is too small, resulting in incomplete picture display; The author needs to carefully edit the format of the paper.

Moderate editing of English language required

Author Response

(The authors gave the same response as above.)

Reviewer 5 Report

The article's topic is undoubtedly relevant and of scientific and practical interest, since in the era of the big data of various nature, methods are needed to increase the efficiency of their processing, adapting in accordance with the changing needs of users, the development of technical means and the capabilities of computer technology. In the abstract, the authors give the essence of the article, briefly describe the research methods, results. The article title and keywords adequately reflect the content of the article.

In the introduction, the authors provide a brief overview of research on the article topic, as well as article structure. The second section is devoted to presenting the theoretical basis of the concepts involved in the development of this study. In the third section, the authors detail previous studies on the perception of visual variables and the analysis of partial overlap of elements. The fourth section presents the methodological decisions and protocols used in setting up and executing the tests in this study. The fifth section presents the initial and final results obtained and describes the collected data. The Discussion section summarizes the main findings and highlights recurring comments from participants. The final section summarizes the results obtained, contains final thoughts on this study, and lists some possibilities for future work.

The article is prepared in accordance with the instructions for the authors, corresponds to the topic that it explores and publishes. Theoretical and practical conclusions are supported by figures and tables that are of sufficient quality. The list of literary sources is adequate to the subject of research.

In our opinion, the article fits the theme of “improving the quality of perception of visual variables for categorical data” and is similar in type to the Preliminary Study.

Comment.

1. In our opinion, it is necessary to more clearly formulate the conclusions, including whether the goal of the study was achieved and by what criteria it was assessed. In our opinion, it is necessary to more clearly define the groups of users who will be able to apply the results of the study.

2. Although the references include links on a given topic, but, in our opinion, it needs to be expanded, since it contains few scientific articles on the topic written in recent years. This is necessary because the industry is dynamically developing.

3. It is necessary to increase the quality of the drawings (for example, up to 240 dpi), or the font size - some symbols are hard to read.

Author Response

(The authors gave the same response as above.)
